# Pre-Launch Exploration of Consumer Willingness to Purchase Selenium- and Iodine-Biofortified Apples—A Discrete Choice Analysis of Possible Market Settings

**DOI:** 10.3390/nu13051625

**Published:** 2021-05-12

**Authors:** Ruth Kleine-Kalmer, Adriano Profeta, Diemo Daum, Ulrich Enneking

**Affiliations:** 1Faculty of Agricultural Sciences and Landscape Architecture, Osnabrück University of Applied Sciences, Am Krümpel 31, 49090 Osnabrück, Germany; d.daum@hs-osnabrueck.de (D.D.); u.enneking@hs-osnabrueck.de (U.E.); 2DIL German Institute of Food Technology, Prof. von Klitzing Str. 7, 49610 Quakenbrück, Germany; a.profeta@dil-ev.de

**Keywords:** biofortification, discrete choice, fruits, health claims, micronutrients

## Abstract

Selenium and iodine are essential micronutrients for humans. They are often deficient in food supply due to low phytoavailable concentrations in soil. Agronomic biofortification of food crops is one approach to overcome micronutrient malnutrition. This study focused on a pre-launch exploration of German consumers’ willingness to purchase selenium- and/or iodine-biofortified apples. For this purpose, an online survey was carried out. In this context, consumers were asked to choose their most preferred apple product from a set card of product alternatives in a discrete choice experiment (DCE). The multinomial logit model results demonstrated that German consumers’ have a particular preference for iodine-biofortified apples. Furthermore, apple choice was mainly influenced by price, health claims, and plastic-free packaging material. Viewed individually, selenium did not exert an effect on product choice whereas positive interactions between both micronutrients exist.

## 1. Introduction

Selenium and iodine are essential micronutrients for human health. They are responsible for the proper functioning of the thyroid and the immune system as well as for the prevention of cancer [1,2,3]. In large parts of Europe and other regions of the world, the selenium and iodine content in soils of agricultural land is relatively low [4,5,6,7]. As a consequence, the population exhibits deficiencies of the micronutrient due to the likewise low selenium content in agricultural products [2]. Iodine deficiency is a serious problem in Germany. Approximately one third of adults and children do not take in sufficient iodine [8,9]. Different measures have been taken in order to prevent the population from the resulting diseases. For example, table salt is often enriched with iodine and made available in supermarkets [10]. In contrast, a similar product containing selenium is not available thus far. However, the reference values for the daily intake of selenium are 60 µg/day for women and 70 µg/day for men in the DACH region (Germany, Austria, and Switzerland) [11]. Different agricultural practices have been applied that increased the supply of micronutrients in food. Examples include the supplementation of feed for livestock with the micronutrients selenium and iodine, resulting in increased levels of these micronutrients in meat and milk, respectively [12].

Another approach is the agronomic biofortification of crops where micronutrients are enriched in plants via fertilization. This technique proved to be effective in increasing the selenium and iodine content in plant-based food [13,14]. Recently, field experiments have been successfully carried out for apples and other fruits [15,16,17,18]. D’Amato et al. compiled numerous attempts of biofortification of food crops with selenium. The authors highlighted that biofortified selenium is well bioavailable for the human body due to its incorporation into organic compounds such as amino acids [19]. In addition, the biofortification of food crops may provide other benefits. For example, in experiments with pears and peaches, foliar selenium sprays resulted in increased soluble solids content and firmness of the fruits, respectively [20]. Furthermore, both selenium and iodine fertilization can promote the formation of valuable plant substances such as vitamin C and phenolic compounds [21,22].

Biofortified crops serve as an example for natural functional food products. Here, the micronutrients were introduced during cultivation instead of processing [21]. Generally, functional foods include ingredients that provide specific health benefits in addition to their natural energy contribution [23]. Apart from serving as meals, their specific features can function as a source of micronutrients that is beneficial for human health [23,24]. Particularly in central and northern European countries, there is an increasing acceptance of functional foods that is mainly due to a change in health consciousness [25]. While a number of trials have been conducted on biofortification with selenium and iodine, only a few food products are currently available in Europe’s food retail market for consumers. One exception is Finland, where the use of mineral fertilizers containing selenium became mandatory in the mid-1980s. This measure has improved the selenium supply of the Finnish population from a very poor status to a sustainably optimal level [14].

In a German consumer survey, Wortmann et al. examined whether fruits like apples were preferred over food supplements as a source of selenium by consumers. They found a higher acceptance for apples that were rich in selenium than for the corresponding food supplements [26]. Apples are among the most popular fruits for Germans [27]. They are also the most often cultivated crops within the fruit category in agriculture [28]. In addition, due to various food scandals in recent years, consumers have a higher trust in fruits that have been grown locally [29]. For this reason, selenium-rich apples were seen as a suitable food alternative to reach large parts of the German population by Wortmann et al. [26].

To embed this new research area in the existing literature, we found studies with analogies to the consumer acceptance of biofortified products. Wortmann et al. provided the first ideas of the general consumer’s acceptance of biofortified apples, but they did not analyze the impact of product attributes in the framework of a purchase decision [26]. Another research project focused on consumers’ willingness to pay for biofortified iodine vegetables in Africa [30]. Furthermore, consumer motivation to consume selenium-rich foods have been examined in Australia [31]. However, none of the two studies focused on apples or on German consumers. In contrast, the intake of the micronutrients selenium and iodine in the form of food supplements, but not biofortified apples, has been analyzed for German consumers [32].

Focusing on ordinary apples not regarding micronutrient content or biofortification, the existing literature of consumer behavior has focused on the influence of apple varieties and the related attributes like taste, sweetness, firmness, or crispiness [33,34,35]. In other studies, consumer preferences were measured by analyzing the impact of quality, price, reduced pesticide input, and packaging as well as organic or local origin [29,36]. In recent years, increasing attention has also been given to ecological packaging in Germany [37], in particular, plastic-free wrapping material for fresh food has become more important [38]. In order to inform consumers about the benefits of foods, producers often use authorized health claims [39]. These support the consumer’s choices by suggesting appropriate products with respect to personal needs. Moreover, health claims often serve as marketing instruments because their beneficial contents add value to the original carrier products [40].

The literature indicated that the consumers’ purchase decision on apples depended on a number of product attributes. Prior to the market launch of the new developed biofortified selenium and iodine apples, the impact of these attributes had not been explored and therefore constituted the main focus in this research. However, biofortified apples are not common products on the food market and German consumers were not familiar with them. The biofortification technique was too complex to explain in the supermarkets. Therefore, a consumer-friendly and market realistic presentation of the benefits of biofortified apples had to be developed. The central approach was communicating the health-related advantages. The two micronutrients and their impact on the body were declared in the form of health-claims and health-related slogans. In order to find out what influence traditional factors plus the new health-related benefits had on the purchase decision of consumers, a DCE designed as a realistic apple purchase situation was performed. Here, different product attributes were experimentally varied and statistically estimated. To sum up, we discuss a new horticultural approach to develop healthy apples that contribute essential and yet incomplete micronutrients to the German population. In addition, this research included an analysis of the consumer acceptance of newly developed biofortified selenium and iodine apples that have never before been tested in a choice experiment designed as a realistic market setting.

To our knowledge, the German consumers’ willingness to purchase selenium- and iodine-biofortified apples has not been examined thus far. Individual product attributes that influence the purchase decisions of consumers in the supermarkets have not been analyzed for German consumers in the context of a shopping situation. These products have not been produced on a larger scale for retail yet. For this reason, the objective of this study was to close the gap between innovative aspects of the new biofortified apples that were ready for market entry on the one hand, and consumer preferences on the other. In the existing literature, no comparable research approaches could be found. This study opens a new, interdisciplinary field of research within the context of biofortification experiments with apples and consumer’s willingness to purchase new biofortified apple products.

## 2. Methods and Material

### 2.1. Data Collection and Survey Design

Data were collected via online interviews in August and September 2019. A fully structured questionnaire was developed. The questionnaire included screening questions regarding the weekly purchase of fresh food, frequency of apple consumption, age, gender, and region. Only participants eighteen years and above who consumed apples at least once a month were allowed to participate, so only people doing the grocery shopping were included. In addition, participants were asked about their most often purchased apple varieties, their present consumption of food supplements, and their knowledge of micronutrients. Furthermore, the questionnaire included a concept description of biofortified selenium and iodine apples. The new biofortified apples with high contents of micronutrients were described and a picture of apple trees was shown for the first time. After the concept description, the DCE followed. More detailed information regarding the experimental design of the DCE is provided in Section 2.3. After respondents were asked to choose their preferred selenium and iodine apple product, a question including statements of further purchase motives of apples was attached. More detailed information regarding purchase motives is provided in Section 2.4 and in question 22 (see Appendix A). The reason for asking this question was to examine whether further product characteristics would affect the purchase decision of biofortified apples. The respondents were recruited and pre-screened for the main responsibility of household shopping by the respondi AG, who provided an online household panel for consumer research. The questionnaire was scripted using LimeSurvey software version 2.00. The collected data were prepared using the statistical software SPSS version 26 and Stata 15.1. χ^2^-tests were applied to analyze the differences between male and female response behavior. Factor analysis and multinomial logistic regression were carried out for more detailed statistical analysis. Data quality checks were implemented regarding the exclusion of incomplete interviews and interviews with a very short interview length. Interviews that failed the question for the level of attention were also excluded from the sample. Overall, *n* = 1042 complete interviews with German consumers accepted for the analysis.

### 2.2. Theoretical Background

DCE is a standard method in consumer research to measure the preferences for new food products [41]. With regard to the research objective, the consumer’s tradeoff between individual levels of attributes of the new selenium and iodine apple products in the purchase decision was analyzed. Consumers were asked to choose one apple product out of three from a set of offered alternatives. The estimated values from all choices are called utility values [42]. The framework for the analysis is called random utility theory and is based on McFadden’s approach with an individual utility function for each consumer depicted in Equation (1) [43,44,45].
(1)$$U_{in}=V_{in}+\varepsilon_{in}$$

According to the random utility framework, each individual *n* has an unobservable utility *U_in_*, which consists of an explainable share *V_in_* and a random component *ε_in_* of utility that is associated with choice alternative *i*. Assuming that *A* is the set of all choice alternatives with *J* elements, then the individual *n* will choose option *i* over another option *j* if the following term applies:(2)$$U_{i}>U_{j}\;where\;i\neq j\;\;\in A$$

That is to say, the utility of option *i* is larger than the utility of option *j*. Equation (3) shows the probability that individual n will choose option *i*:(3)$$P_{in}=P\;\left[\left\{\varepsilon_{jn}-\varepsilon_{in}\right\}<\left\{V_{in}-V_{jn}\right\}\right]\;for\;i\neq j\;\;\in A$$

For the calculation of the choice probabilities, the distribution of the random component *ε_in_* needs further specification based on McFadden’s basic approach. Equation (4) shows the multinomial conditional logit model that is often applied in DCE:(4)$$P_{in}=\frac{e^{V_{in}}}{\sum_{j=1}^{J}e^{V_{jn}}},\;j=1,\;\ldots ,\;J\;and\;j\neq i$$

In order to calculate the multinomial conditional logit model in this research, attributes and levels of the biofortified apples had to be determined.

### 2.3. Design of the Discrete Choice Experiment

From the outside, biofortified apples do not differ from other apples and consumers are not able to distinguish between ordinary and biofortified apples. For the illustration in the online questionnaire, realistic apple products were needed. Several informal interviews were first conducted with producers of apples and scientific experts in the field of plant research. Based on their expertise and the above-mentioned criteria, the attributes in this DCE were developed. In order to demonstrate the specifics of the new biofortified apples to consumers, packaged apples were chosen over loose ones. The advantage of this form was that product information could be placed directly on the apples. For the DCE, choice sets were created as choice cards including varying product attribute combinations. They represented ordinary apple products that were available in German supermarkets in June 2019 based on market observations before the survey was conducted. Thereby, the focus of the respondents was not directly pointed at the novelty (the micronutrients selenium and iodine), but on the apple product as a whole. In order to find out which variables offered the highest utility in the choice of biofortified apples, the following questions were examined:Q1: Which micronutrient combination is most preferable?Q2: Which is the most preferable health-related slogan?Q3: Which is the most preferable variety?Q4: Which is the most preferable packaging material and format?Q5: Will consumers of selenium and iodine food supplements have a higher acceptance of biofortified apples than non-consumers of food supplements?

Q1 focused on the two micronutrients selenium and iodine. They have not been offered to potential German consumers in form of biofortified apples yet. It was crucial to examine whether consumers preferred selenium or iodine or both micronutrients in combination. Additionally, one option without micronutrients was tested. This was the most realistic option for German consumers at that time because they had neither heard about biofortified apples nor seen them in retail yet.

Q2 focused on the information regarding the benefits of the new biofortified apples. Respondents were provided with basic background knowledge of the benefits of biofortified apples by testing the health claims “Contributes to the normal function of the immune system” and “Contributes to the normal production of thyroid hormones and normal thyroid function” [46]. In addition, one slogan describing the content of micronutrients and one slogan about the enjoyment of eating healthy and delicious apples were tested. It was crucial to find out which of the offered health-related slogans offered the highest utility to consumers.

In Q3, the most preferred varieties of the consumers were examined because previous research regarding the choice of apples has shown that varieties played an important role for consumers in the purchase decision. Varieties were associated with the preferred taste and had a positive effect on the choice of apples. In particular, club apples like Pink Lady^®^ became more important [47]. The most preferred varieties by German consumers Elstar, Braeburn, and Pink Lady^®^ were included in the DCE as well as a new developed brand named Selstar^®^ [48].

Due to increasing awareness of sustainable purchasing behavior, consumers are paying more and more attention to the consequences of their shopping behavior. By choosing plastic-free packaging material, an increasing consumer segment contributes to the reduction of microplastics [37]. Additionally, the general reduction of waste and efforts that are required for recycling packaging material are coming to the fore [38]. Q4 consequently investigated the packaging material to find out if it affected the consumers’ decision in the context of biofortified apples.

With regard to target groups for the new biofortified apples, the segment of buyers of food supplements was identified. The group revealed a higher acceptance for biofortified apples in previous research, mainly because they were familiar with food supplements to obtain particular micronutrients [26]. For this reason, the fifth question addressed potential groups of buyers to find out if the consumption of selenium and iodine food supplements affected the willingness to purchase biofortified apples positively.

To sum up, the attributes’ packaging, health-related slogan, variety, “rich in” (micronutrients), and price were selected for the DCE (Table 1). The price levels were based on market observations in supermarkets. Overall, the DCE consisted of five attributes with four levels, except for packaging, which only had three levels.

Regarding the different formats of packaging, pictures were shown to the respondents in the choice sets. This supported the visual imagination of the new apple products in the online questionnaire. Since the packaging constituted the outer appearance of the new biofortified apples, it also had an impact on the initial impression of the potential consumers. For that reason, three types of packaging were used as labels (fixed label design) in the DCE and therefore did not vary on the set cards. The fixed label design was also taken into consideration later in the regression analysis. Packaging A comprised plain cardboard without imprints, which partly covered the apples. Packaging B was a plastic bowl with transparent plastic foil where the apples were completely covered but visible through the foil. Packaging C comprised of color printed cardboard that covered most of the fruits. Below the pictures of the packaging was a banner including the health-related slogans. The prices were declared in Euros and referred to the entire product unit. Overall, the different apple products were always shown as version A, B, and C in one choice set with fixed ordering of packaging and varying health-related slogans, varieties/brands, micronutrients, and prices (Figure 1). Before the respondents were asked to make a choice from the displayed choice sets, the concept of the new biofortified apples was introduced at the beginning. The description was developed according to merchantable commercial material that briefly described the presence of both micronutrients and their benefits. The following text was shown to the respondents (compare section concept test and Q21 in Appendix A):

“In the next section, we would like to introduce to you a new kind of apple which is especially rich in selenium and iodine. Selenium and iodine are essential micronutrients which contribute significantly to the improvement of human health.

The first companies are currently launching the new biofortified apples in retail. Please imagine you are in a supermarket and would like to purchase a package of 6 apples. Which of the following apples would you choose?”

Overall, twenty choice sets were developed and randomly distributed among the sample (Appendix A
Table A1). Figure 1 shows one example of a choice set.

In order to minimize the number of decisions in the online interview, the number of choice sets was reduced with the software Ngene version 1.2 [49]. Consequently, each respondent was shown one choice set and only had to pick one out of three offered alternatives for apple products. Herewith, an overload of choices, which often caused fatigue in the response behavior was avoided.

Equation (5) describes the final estimated multinomial logit model of the statistical analysis. All experimental variables were included as well as the interaction terms of the micronutrients and the packaging, the interaction of selenium and iodine with food supplements and packaging, and the interaction of selenium and iodine with further motivations for purchasing the new biofortified apples and packaging. The interaction terms with packaging were considered because of the fixed label design of the DCE. The motivations to buy biofortified apples were condensed by means of factor analysis (Section 2.4) beforehand and refer to the additional benefits of the new apples that the respondents had to rate in Q22 of the questionnaire.
(5)$$\begin{array}{cc} V=\beta_{1}*\mathrm{constant} & \mathrm{cardboard}+\beta_{2}*\mathrm{constant}\;\mathrm{color}\;\mathrm{print}+\beta_{3}*\mathrm{price}\;\mathrm{cardboard}+\beta_{4}\\ & *\mathrm{price}\;\mathrm{plastic}+\beta_{5}*\mathrm{price}\;\mathrm{color}\;\mathrm{print}+\beta_{6}*\mathrm{Pink}\;\mathrm{Lady}{}^{®}+\beta_{7}\\ & *\mathrm{Elstar}+\beta_{8}*\mathrm{Braeburn}+\beta_{9}\\ & *\mathrm{Contributes}\;\mathrm{to}\;a\;\mathrm{normal}\;\mathrm{functioning}\;\mathrm{of}\;\mathrm{the}\;\mathrm{immune}\;\mathrm{system}+\beta_{10}\\ & *\mathrm{Tasty}\;\mathrm{and}\;\mathrm{healthy}\;\mathrm{enjoyment}+\beta_{11}\\ & *\mathrm{Naturally}\;\mathrm{with}\;\mathrm{micronutrients}+\beta_{12}*\mathrm{Selenium}\;x\;\mathrm{cardboard}+\beta_{13}\\ & *\mathrm{Selenium}\;x\;\mathrm{plastic}+\beta_{14}*\mathrm{Selenium}\;x\;\mathrm{color}\;\mathrm{print}\;+\beta_{15}\\ & *\mathrm{Iodine}\;x\;\mathrm{cardboard}+\beta_{16}*\mathrm{Iodine}\;x\;\mathrm{plastic}+\beta_{17}\\ & *\mathrm{Iodine}\;x\;\mathrm{color}\;\mathrm{print}+\beta_{18}*\mathrm{selenium}\;\&\;\mathrm{iodine}\;x\;\mathrm{cardboard}+\beta_{19}\\ & *\mathrm{selenium}\;\&\;\mathrm{iodine}\;x\;\mathrm{plastic}+\beta_{20}*\mathrm{selenium}\;\&\;\mathrm{iodine}\;x\;\mathrm{color}\;\mathrm{print}+\beta_{21}\\ & *\mathrm{selenium}\;\&\;\mathrm{iodine}\;x\;\mathrm{food}\;\mathrm{supplement}\;x\;\mathrm{cardboard}+\beta_{22}\\ & *\mathrm{selenium}\;\&\;\mathrm{iodine}\;x\;\mathrm{food}\;\mathrm{supplements}\;x\;\mathrm{plastic}+\beta_{23}\\ & *\mathrm{selenium}\;\&\;\mathrm{iodine}\;x\;\mathrm{food}\;\mathrm{supplements}\;x\;\mathrm{color}\;\mathrm{print}+\beta_{24}\\ & *\mathrm{selenium}\;\&\;\mathrm{iodine}\;x\;\mathrm{information}\;\mathrm{requirement}\;x\;\mathrm{cardboard}+\beta_{25}\\ & *\mathrm{selenium}\;\&\;\mathrm{iodine}\;x\;\mathrm{practicability}\;x\;\mathrm{cardboard}+\beta_{26}\\ & *\mathrm{selenium}\;\&\;\mathrm{iodine}\;x\;\mathrm{sustainability}\;x\;\mathrm{cardboard}+\beta_{27}\\ & *\mathrm{selenium}\;\&\;\mathrm{iodine}\;x\;\mathrm{information}\;\mathrm{requirement}\;x\;\mathrm{plastic}+\beta_{28}\\ & *\mathrm{selenium}\;\&\;\mathrm{iodine}\;x\;\mathrm{practicability}\;x\;\mathrm{plastic}+\beta_{29}\\ & *\mathrm{selenium}\;\&\;\mathrm{iodine}\;x\;\mathrm{sustainability}\;x\;\mathrm{plastic}+\beta_{30}\\ & *\mathrm{selenium}\;\&\;\mathrm{iodine}\;x\;\mathrm{information}\;\mathrm{requirement}\;x\;\mathrm{color}\;\mathrm{print}+\beta_{31}\\ & *\mathrm{selenium}\;\&\;\mathrm{iodine}\;x\;\mathrm{practicability}\;x\;\mathrm{color}\;\mathrm{print}+\beta_{32}\\ & *\mathrm{selenium}\;\&\;\mathrm{iodine}\;x\;\mathrm{sustainability}\;x\;\mathrm{color}\;\mathrm{print} \end{array}$$

In this discrete choice study, the estimated *β* coefficients indicate how the respective variable influences the choice of one of the three offered alternatives. The alternatives correspond to the three packaging formats. The corresponding *p* values show whether the observed variable influences the probability of choosing the respective alternative significantly (three significance levels: *** *p* ≤ 0.001; ** *p* ≤ 0.01, * *p* ≤ 0.05).

### 2.4. Factor Analysis

Factor analysis was applied in order to further examine the purchase motivations of apple products, which were integrated into the multinomial conditional logit model (Equation (5)). It is a common method applied in marketing research for the purpose of discovering invisible components that affect consumers’ decisions. Steptoe et al. extracted nine factors as basic motives for selecting food. Among them were convenience, sensory-appeal, natural content, familiarity, or similar motivations [50]. A question regarding motives for purchasing biofortified apples was also developed for this research. Pre-formulated items and interval-scaled responses were included in the questionnaire. The respondents were asked to rate statements on a 5-point Likert-scale that ranged from “Is very appealing to me” to “Is not appealing to me at all”. The terms were formulated as additional benefits of the new biofortified apples and reasons for purchasing them. Included were descriptions regarding the origin of the apples, sensory-appeal of the fruits, packaging specifics, and information regarding the production. Interviews including missing values were excluded from the factor analysis. The extracted factors were included in the discrete choice model.

## 3. Results

### 3.1. Sample Description and Socio-Demographic Criteria

After data preparation and quality checks, the final sample included a balanced distribution of gender with 51.4% female and 48.6% male respondents, as displayed in Table 2. About one third of the respondents were older than 54 years (32.0%) and most of them lived in the densely populated western part of Germany (35.4%), followed by the south with 27.6%. Regarding the respondents’ occupation, more than half were employed full-time (56.5%), followed by 13.4% part-time, and 12.1% were retired. Generally, the level of employment was rather high. Regarding the educational level, 35.7% held a university degree. Considering household size, most interviewees lived in a household with two people (38.2%) and 27.8% lived in a single household. Only very few respondents lived in households with five or more people. A total of 19.0% of all respondents had children under twelve years.

### 3.2. Consumption Habits of Apples and Food Supplements

About a quarter of the respondents (24.1%) ate apples on a daily basis and another 34.5% consumed apples several times a week. Overall, apples were part of the regular diet of most respondents and only a minor segment (8.0%) of the respondents indicated eating apples only rarely. However, it must be noted that apple consumption was a mandatory screening criterion so that only people who ate apples at least once per month were considered for the survey. In the older age groups, there was a larger share of respondents who ate apples on a daily basis. A total 32.4% of consumers who were older than 54 years ate apples daily compared to 14.4% of consumers aged 18–24 years. For the varieties, the results showed that Pink Lady^®^ was mentioned as the most often purchased variety with a share of 23.8% (Table 3). It was also the most often purchased variety by women with 25.9% and men with 21.5%. Braeburn ranked second with 12.6% of all mentions. However, looking at the differences between men and women, it can be noted that whereas women ranked Braeburn second with 14.4%, Golden Delicious was the second most often purchased variety by men with 14.7%. Only 10.7% of men chose Braeburn as their most often purchased variety. Therefore, in the case of Braeburn, a difference in the purchasing behavior between men and women was recorded. The third most often purchased variety was Elstar, with 12.0% of all mentions and 11.0% of men and 13.1% of women. As depicted in Table 3, Pink Lady^®^, Braeburn, and Elstar were the most often purchased varieties for 48.4% of German consumers.

Regarding the knowledge about the two micronutrients iodine and selenium, consumers were asked to choose only those micronutrients from a list that they had heard of before. Generally, the overall awareness of iodine (71.3%) was much higher than the awareness of selenium (46.9%) (see Table 4). Furthermore, there was a difference in the awareness of the two micronutrients between men and women. A total of 76.3% of all women were aware of iodine and in contrast, 66.0% of all male respondents had ever heard about iodine (χ^2^: *p* ≤ 0.001). In the case of selenium, the female share was 51.9% and the share of men was only 41.7% (χ^2^: *p* = 0.001). In addition, the respondents were asked to choose those micronutrients from a list that they had previously consumed in the form of food supplements. A difference in the consumption between selenium and iodine was found. A total of 19.3% of all respondents had consumed iodine in the form of food supplements and 12.5% had consumed selenium in the form of food supplements.

In the consumption patterns, a difference between men and women could be observed. A total of 21.5% of the women declared that they had consumed iodine as a food supplement compared to 17.0% of men. However, the difference was not significant (χ^2^: *p* = 0.068). In the case of selenium, the difference was significant and higher with 16.4% of women who had consumed selenium as a food supplement and 8.3% of men (χ^2^: *p* = 0.000). Hence, iodine was the more common micronutrient and was also more often consumed as a food supplement than selenium. Additionally, women were more often aware of both micronutrients and consumed both of them more often in the form of food supplements. 

### 3.3. Results of the Factor Analysis

Only items with factor loadings with values larger than 0.5 were displayed (Table 5). The extracted factors explained 59.1% of the variation and were allocated to three different factors. Therefore, it was assumed that the extracted components contributed well to explaining the measured variance of the purchase motives of the new biofortified apples.

The three extracted factors can be described as follows:

Factor 1: This was called sustainability because it encompassed regional, seasonal, and freshness aspects of the apples. It consisted of statements with high factor loadings addressing the country of origin (Germany) and the region of production of the apples, which was close to the respondents’ homes. Additionally, it included a statement regarding organic cultivation. Moreover, the time of harvest loads highly on this factor as well as the statement regarding the plastic-free packaging material. Intense fruity taste and integrated production are the last two terms that loaded on the first component.

Factor 2: This was called practicability and included ideas regarding a long shelf life, intense red fruit husk, and an aesthetically appealing packaging. In addition, the statement regarding the easy opening of the packaging loaded highly on this factor. Finally, the statement regarding the well-known and third most often purchased variety ‘Elstar’ loaded highly on this factor. This is called practicability because mainly items regarding reduced efforts in the handling of apples and simplifying choice loaded on this factor.

Factor 3: This was called information requirement and comprised additional information regarding suitability for vegan and vegetarian nutrition. Additionally, the certification of the University Medical Center Charité Berlin loaded highly on this factor. One term regarding the availability of further information on the Internet and the information regarding the suitability for low allergies nutrition were also included in this factor. Generally, the factor comprised ideas regarding specific information and safety and spent details on the product.

### 3.4. Results of the Discrete Choice Analysis

In the DCE-model, most of the analyzed parameters revealed positive effects on the apple choice except for the price, which exerted a negative effect. However, the intensity of the effects differed considerably (Table 6). For the micronutrients, the findings showed that selenium did not have a significant effect on the willingness to purchase apples. None of the coefficients of the interaction terms of selenium and the packaging were significant. In contrast, the interaction coefficient of iodine and the cardboard packaging (0.64 *) as well as iodine and the plastic container (0.68 *) were significant, whereas only the coefficient of the color print packaging was not significant. However, the interaction of both micronutrients and the color print packaging had a significant positive effect. Remarkably, the coefficient was 0.73 * and had a higher effect than in the case of the interaction of iodine and the other two packaging variants. With the color print packaging, the combination of both micronutrients had a stronger effect than each micronutrient individually.

The results also showed that the health claim regarding the normal functioning of the immune system had the strongest impact (0.80 ***) on the willingness to purchase the new biofortified apples. The slogans “Tasty and healthy enjoyment” (0.51 ***) and “Naturally with micronutrients” (0.54 **) were also significant and positive. Concerning the varieties, Pink Lady^®^ (0.60 ***) and Elstar (0.40 **) both had a significant and positive effect whereas neither Braeburn nor Selstar^®^ showed such effect. The interaction term of both micronutrients, cardboard container, and information requirement was 0.45 ** and had a significant positive effect. However, the effect of the interaction of both micronutrients, cardboard container, and sustainability was stronger with a positive and significant coefficient of 0.64 ***. In the interaction of both micronutrients, plastic packaging, and purchase motives, there was only one statistically significant negative coefficient of −0.38 ** in the case of sustainability. Color print packaging, both micronutrients and purchase motives did not show significant coefficients. The interaction terms of both micronutrients, the consumption of selenium and iodine as food supplements, and the packaging did not have any effects either. In contrast to all the other variables, the price consistently had a statistically negative effect on the willingness to purchase for all packaging formats. The strongest negative effect of the price was measured for the cardboard packaging with a coefficient of −1.16 ***, followed by −0.68 ** in the case of plastic, and −0.65 ** in the case of the color print. 

## 4. Discussion

### 4.1. Influence of Presence of Selenium and Iodine

In the DCE, the micronutrient selenium alone did not have a significant effect on the willingness to purchase biofortified apples. One explanation for this could be the limited knowledge about selenium. Only 46.9% of the respondents had ever heard of selenium (Table 4). Considering the effect of iodine alone, the results of the discrete choice model turned out to be significant and positive. The consumers were more familiar with this micronutrient (71.3%). Additionally, due to established functional foods like iodized salt, the insufficient supply of iodine as well as the presence of iodized food products in Germany was more commonly known [51]. This means that the majority of the respondents were also aware of the relevance of iodine for human health. The combination of both micronutrients affected the willingness to purchase the color print positively. Likewise, the effect here had the highest coefficient. On the one hand, this effect could be caused by the positive effects of iodine. On the other hand, this effect could also be carried by the assumption that the positive effect of iodine for health is reinforced by the presence of selenium. Possibly, the lack of knowledge about selenium and its positive effects on human health were counterproductive in the decision making process. With reference to Q1, it can be noted that iodine and both micronutrients combined had a positive effect. In order to promote the micronutrient selenium, further details about selenium deficiencies and its health benefits need to be provided with the biofortified apples. Preferences for special product types (color print) should be considered.

### 4.2. Influence of Health-Related Slogans

The health-related slogans encompassed four different levels including two authorized health claims. The latter explained the benefits of selenium and iodine. The other two levels addressed, in addition to health aspects, the taste of the new biofortified apples and the natural content of micronutrients. Previous research has shown that physiological health claims would have a higher value if they were combined with a healthy carrier [52]. Similarly, the present results confirm a positive effect of health-related slogans on the willingness to purchase new biofortified apples. The results were thus also in line with previous studies that examined the importance of health claims [25,40,53,54]. In the case of selenium and iodine apples, consumers had a positive attitude toward the health claim regarding the normal functioning of the immune system. In contrast, the second health claim regarding normal thyroid function did not have a positive effect on the willingness to purchase. Furthermore, the slogans addressing the taste and the natural content of micronutrients of the apples had positive effects. Consequently, with regard to Q2, the health claim concerning the normal functioning of the immune system had the strongest effect on the willingness to purchase the new biofortifed apples. This provided sufficient background information about the innovative aspects of the new biofortified apples and stressed the additional benefits for the consumers.

### 4.3. Influence of Variety and Brand Name

In previous apple choice experiments conducted by Yue and Tong, varieties were associated with specific product characteristics such as taste or texture and were recognized by the consumers [35]. Since apple tastings could not be provided in our online approach, two varieties and two brand names for apples were included. Elstar and Braeburn were among the most often purchased apple varieties in Germany (Table 3). Pink Lady^®^ was the most often purchased variety by 23.8% of the respondents. Since there had not been a comparable selenium- or iodine-rich apple in German retail previously, a brand name for a biofortified selenium apple was also included. It was called Selstar^®^ and was shown to the consumers in the DCE for the first time [48]. The results showed that the two most common varieties Pink Lady^®^ and Elstar had positive effects on the willingness to purchase the new biofortified apples. This was in line with previous research findings where variety had a positive effect and price a negative effect on the willingness to purchase apples [35,55]. This means that concerning Q3 and the variety, the German consumers acted as expected. Looking at the brand name Selstar^®^, which represented a new name for selenium-rich apples, no effects on the willingness to purchase were found. Keeping in mind that the respondents had only just learned about this new brand name, the finding was not surprising. Since none of the consumers had ever heard of the name Selstar^®^ nor seen it in German retail, none of the respondents could refer to a reference of this product. Among the three well established apple varieties and brands in the German market, the new name was not recognized. To sum up, the most common variety Pink Lady^®^ had the strongest effect among the varieties or brand names.

### 4.4. Influence of Additional Purchase Motives

The analysis of additional purchase motives in the DCE has shown that a positive effect on the willingness to purchase was found if the factor sustainability interacted with cardboard material and both jointly displayed micronutrients (0.64 ***) were considered and a negative effect if interacted with plastic packaging (−0.38 **). These findings were in line with current research findings regarding a negative attitude of German consumers toward plastic packaging. Recently, even a trend toward zero-packaging material could be observed. The prohibition of plastic bags in German retail was one example of the recent political decisions [56]. Consequently, the traditional plastic packaging B did not have positive effects on the willingness to purchase the new biofortified apples. 

Apart from that, positive effects of the interaction of both micronutrients, the factor information requirement, and the cardboard packaging also reflected a recent trend in consumer behavior in Germany. With regard to production and processing methods and origin of food, consumers have become increasingly concerned [57]. For this reason, producers have increased their attempts to provide more detailed information regarding the supply and production chains of their products. In this manner, they have tried to overcome mistrust on the consumer side. Additionally, consumers tended to decide in favor of locally produced food products as they assumed a higher food safety [58]. 

The factor practicability, in contrast, did not have a significant effect on the willingness to purchase the new biofortified apples, regardless of the packaging material. It seemed that the benefits of easy handling and little efforts did not matter to a larger extent. However, the two factors, sustainability and information requirements, showed positive effects. In this DCE, a tendency toward more critical reflection of product information and environmentally friendly packaging material could be observed. With regard to Q5, plastic-free packaging material was found to be the preferred packaging in this DCE.

### 4.5. Influence of Consumption of Food Supplements

Previous research has demonstrated that the consumption of food supplements had increased in recent years [32]. For a lot of people, food supplements settled in the regular nutrition, and from the perspective of elderly people, they were considered as important as prescribed medicine for health [58]. In this research, 19.3% of the respondents mentioned that they had consumed iodine as a food supplement and 12.5% selenium. However, the idea that these consumers would be more attracted to the biofortified apples could not be verified. The interactions of both micronutrients, consumption of food supplements, and the packaging did not have significant effects. One possible explanation could be that the consumers of food supplements were already provided with sufficient micronutrients. Therefore, they did not need to consume the micronutrients in the form of apples anymore. However, it did not seem that they considered the biofortified selenium and iodine apples as equivalent to food supplements either. Another explanation could be that the apples were still unknown and consumers were not adequately convinced by the information provided in the questionnaire.

### 4.6. Influence of Price

The product price was included in the experiment to give consumers an indication of the costs they would have to spend for the new biofortified apples. The price steps were chosen based on market observations in German retail for similar apple products. Price had a negative impact on the willingness to purchase apples in previous research projects [29]. This was confirmed by the results of this DCE as well as the negative influence of price on the willingness to purchase apples.

### 4.7. Limitations of the Present Study and Implications for Further Research

Overall, the research approach implied the following limitations. The brand name Selstar^®^ was included in the choice sets, but the consumer had never heard of it. No explanation was given beforehand. The pictures of the packaging were supposed to support the visual imagination of the product. However, they contained different colored apples, which could have influenced the respondents. The health-related slogan “Contributes to a normal functioning of the immune system” is not authorized for iodine. However, in the experiment, it was combined with iodine in the choice sets. If the results of this study are transferred to real products in the supermarkets, this constraint of research will have to be taken into consideration by the producers. 

Furthermore, the choice experiment aimed at creating a decision situation, which was close to the shopping reality in German supermarkets. Nevertheless, the respondents were not aware of reference products from real supermarkets because the new biofortified apples were still at the developing stage. Given the challenge to combine specific micronutrient information and usual apple buying behavior knowledge, our approach was limited. It can be considered as a first approach in the field of consumer acceptance analysis with respect to biofortified apples. We could not refer to sophisticated previous research in this field and therefore had to reduce the discrete choice design as well as the statistical model to a manageable extent. Future research could build upon our results and identify more significant interactions between micronutrient information and further drivers of apple preferences, buying behavior, and socio-demographics.

In addition to this research, in a recently performed test launch in more than 300 outlets in Germany, around 50,000 units of selenium-biofortifed apples from the 2020 harvest were sold under the brand name “Selstar^®^”. In an accompanying customer survey (n = 336), almost every second participant (49%) stated that they particularly liked the Selstar’s contribution to healthy nutrition. The majority of buyers (87%) said that they would purchase this apple again, 12% were still undecided in this respect, and only less than 1% were no longer interested in the apple. The results indicate a very high acceptance level of selenium-rich apples among German consumers, which should also be further explored in future studies.

## 5. Conclusions

The DCE was conducted in a German online consumer survey to examine the impact of individual product attributes on the willingness to purchase biofortified apples prior to market launch. Considering the isolated effect of the micronutrient selenium, it was assumed that the benefits derived from the supply of selenium were not sufficiently communicated yet. In the case of iodine, a positive effect was found. If selenium and iodine biofortified apples shall serve as regular, natural selenium, and iodine source for vast parts of the German population in the future, the benefits of the consumption need to become more obvious to all consumers. The intensification of marketing at the point-of-sale could be one solution for that. More information about the consequences of insufficient supply with selenium on human health could be provided in order to increase the awareness for health problems that result from constant deficiencies. Generally, it should become more visible to the German population, where health damage could occur if the insufficient supply of these two essential micronutrients continued. Apart from that, the results of the DCE indicated that consumer purchase decisions were influenced by a set of product attributes. Common varieties and appropriate prices were crucial for the purchase decision of biofortified apples. However, given sufficient background information, the results suggest that micronutrients can also positively influence purchase decisions on apples. Micronutrients, especially those with limited awareness like selenium, should be combined with more well-known examples like iodine, health-related slogans, and relevant marketing attributes in order to make the products more preferable to consumers.

## Figures and Tables

**Figure 1 nutrients-13-01625-f001:**
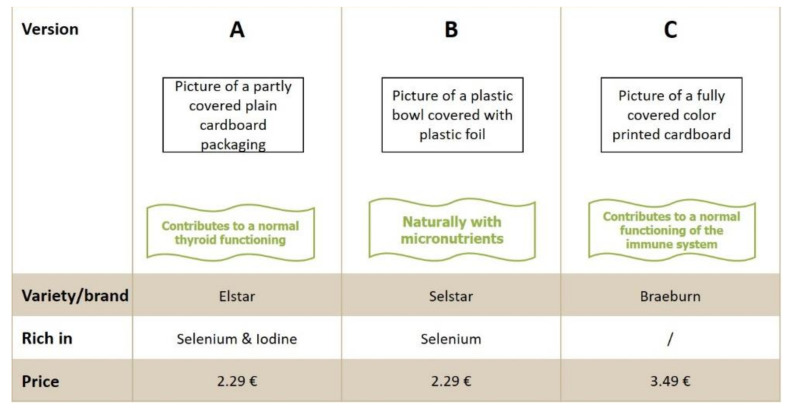
Example of one choice set card with three choices (**A**–**C**) out of a total of twenty set cards with different choice alternatives.

**Table 1 nutrients-13-01625-t001:** Attributes and levels of the DCE.

Attributes	Levels
Packaging ^1^	Cardboard foodtainer, banderoles, no print
	Plastic bowl, plastic foil
	Foodtainer, hardcover slipcase, color print
Health-related slogan	Tasty and healthy enjoyment
	Naturally with micronutrients
	Contributes to a normal functioning of the immune system
	Contributes to a normal thyroid functioning
Variety/brand	Elstar
	Braeburn
	Pink Lady^®^
	Selstar^®^
Micronutrients (rich-in)	Selenium
	Iodine
	Selenium and iodine
	No micronutrients stated on the choice set
Price	2.29 €
	2.49 €
	2.99 €
	3.49 €

^1^ The packaging was not rotated (i.e., remained in the same order).

**Table 2 nutrients-13-01625-t002:** Characteristics of survey participants and social-demographic criteria.

Characteristic	Unit	Respondents	
		n	%
Total		1042	100
Gender	Female	536	51.4
	Male	506	48.6
Age	18–24 years	97	9.3
	25–34 years	195	18.7
	35–44 years	173	16.6
	45–54 years	244	23.4
	>54 years	333	32.0
Region	West	369	35.4
	South	288	27.6
	North	193	18.5
	East	192	18.4
Occupation	Full-time employed	589	56.5
	Part-time employed	140	13.4
	Retired	126	12.1
	Student	74	7.1
	Homemaker	44	4.2
	Unemployed	38	3.7
	Practical education	14	1.3
	School	7	0.7
Education	University degree	372	35.7
	Secondary school	277	26.6
	High school diploma	206	19.8
	Elementary school (Hauptschule)	104	10.0
	Specialist/Master craftsman	76	7.3
	Still attending school	4	0.4
Size of the household	1 Person	290	27.8
	2 Persons	398	38.2
	3 Persons	174	16.7
	4 Persons	127	12.2
	5 Persons	37	3.6
	More than 5 Persons	16	1.5
Parenthood	No children	548	52.6
	Children under twelve years	198	19.0

**Table 3 nutrients-13-01625-t003:** Most often purchased apple varieties of German consumers by gender.

Varieties	Total ^1^		Male		Female	
	n	%	n	%	n	%
Pink Lady^®^	239	23.8	104	21.5	135	25.9
Braeburn	127	12.6	52	10.7	75	14.4
Elstar	121	12.0	53	11.0	68	13.1
Golden Delicious	103	10.3	71	14.7	32	6.1
Jonagold	94	9.4	47	9.7	47	9.0
Granny Smith	68	6.8	32	6.6	36	6.9
Boskoop	61	6.8	29	6.0	32	6.1
Gala	40	4.0	20	4.1	20	3.8
Red Delicious	37	3.7	21	4.3	16	3.1

Comparison male and female respondents: χ^2^: *p* = 0.002; 1 n = 1005; only varieties with more than 3.0% of mentions are displayed.

**Table 4 nutrients-13-01625-t004:** Knowledge of selenium and iodine as well as consumption as a food supplement by gender.

Micronutrients	Status	Total		Male		Female	
		n	%	n	%	n	%
Iodine ***	Stated awareness	743	71.3	334	66.0	409	76.3
	Unaware	299	28.7	172	34.0	127	23.7
Selenium ***	Stated awareness	489	46.9	211	41.7	278	51.9
	Unaware	553	53.1	295	58.3	258	48.1
Iodine as food supplement (n.s.)	Consumers	201	19.3	86	17.0	115	21.5
	Non-consumers	841	80.7	420	83.0	421	78.5
Selenium as food supplement ***	Consumers	130	12.5	42	8.3	88	16.4
	Non-consumers	912	87.5	464	91.7	448	83.6

Comparison male and female respondents: χ^2^: *** *p* ≤ 0.001, n.s. *p* ≥ 0.05.

**Table 5 nutrients-13-01625-t005:** Rotated component matrix of principal component analysis and allocation to factors.

Factors	Statements	Factor Loadings
Sustainability	Produced in Germany	0.84
	Produced locally/nearby	0.82
	Organically grown	0.72
	Freshly picked and crispy	0.69
	Plastic-free packaging	0.66
	Intense fruity taste	0.65
	Integrated production	0.52
Practicability	Warranty of long shelf life fruits	0.72
	Intense red fruit peel	0.70
	Aesthetically appealing packaging	0.69
	Easy to open	0.65
	Handy, medium fruit size	0.57
	Popular apple variety ‘Elstar’	0.52
Information requirements	Especially suitable for vegetarians or vegans	0.78
	Imprint: Certified by University Medical Center Charité Berlin	0.71
	Further information available on the Internet (e.g., via QR code)	0.70
	Especially suitable for low allergy nutrition	0.68

KMO-criterion: 0.914.

**Table 6 nutrients-13-01625-t006:** Results of the multinomial conditional logit model.

	*β* Coefficients	Standard Error
**Constants**		
Constant cardboard	2.01 *	0.96
Constant color print	0.22	1.02
**Interaction terms micronutrients (rich in) and packaging**		
Selenium x cardboard	0.05	0.37
Selenium x plastic	0.13	0.27
Selenium x color print	0.23	0.24
Iodine x cardboard	0.64 *	0.25
Iodine x plastic	0.68 *	0.29
Iodine x color print	0.21	0.29
Selenium & iodine x cardboard	0.48	0.29
Selenium & iodine x plastic	0.48	0.28
Selenium & iodine x color print	0.73 *	0.32
**Nutrition and health claims**		
Contributes to a normal functioning of the immune system	0.80 ***	0.17
Tasty and healthy enjoyment	0.51 ***	0.14
Naturally with micronutrients	0.54 **	0.16
**Varieties**		
Pink Lady^®^	0.60 ***	0.14
Elstar	0.40 **	0.14
Braeburn	0.21	0.14
**Interaction terms micronutrients (rich in), purchase motives, and cardboard packaging**		
Selenium & iodine x information requirement x cardboard	0.45 **	0.16
Selenium & iodine x practicability x cardboard	0.13	0.14
Selenium & iodine x sustainability x cardboard	0.64 ***	0.16
**Interaction terms micronutrients (rich in), purchase motives, and plastic packaging**		
Selenium & iodine x information requirement x plastic	0.25	0.14
Selenium & iodine x practicability x plastic	0.16	0.15
Selenium & iodine x sustainability x plastic	−0.38 **	0.14
**Interaction terms micronutrients (rich-in), purchase motives, and color print**		
Selenium & iodine x information requirement x color print	−0.10	0.15
Selenium & iodine x practicability x color print	0.09	0.15
Selenium & iodine x sustainability x color print	0.13	0.15
**Interaction terms micronutrients, food supplements, and packaging**		
Selenium & iodine x food supplement x cardboard	−0.12	0.31
Selenium & iodine x food supplements x plastic	−0.06	0.33
Selenium & iodine x food supplements x color print	0.50	0.35
**Price**		
Price cardboard	−1.16 ***	0.24
Price plastic	−0.68 **	0.21
Price color print	−0.65 **	0.21

Pseudo R^2^: 0.11; Significance level *** *p* ≤ 0.001; ** *p* ≤ 0.01, * *p* ≤ 0.05.

## Data Availability

The data presented in this study are available on request from the corresponding author. The data will be made publicly to a later stage.

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
