# Peer review of "Pre-Launch Exploration of Consumer Willingness to Purchase Selenium- and Iodine-Biofortified Apples—A Discrete Choice Analysis of Possible Market Settings"

_nutrients, 2021, doi:10.3390/nu13051625_

Round 1
Reviewer 1 Report
Comments are included in the file attached.

Reviewer 2 Report
The manuscript evaluates the novel approach to food fortification, namely biofortification. In the particular objective the survey study explores the necessary attributes to make new appearing apple varieties to be more appealing for the consumers.
The biofortified varieties are not common on the German market and the authors give the evidence that this is an important factor of choice. It is a relevant evaluation if it shows what could be done prior to market launch of this biofortified products when supported with their claimed benefits. Some hypothesis are vey obvious and the questions was why to test them, while one of the is very important and refers to choice of food supplement above biofortified apples. This one has been shown as a wrongly set. This knowledge should be further utilized and explored.
The study is designed trough a set of hypothesis, choice set and online survey. The survey design should be described with more indication of the purpose, length and the set of questions, choice of respondents and the season when the survey was done. On the other hand the evaluation of the study results is explained in more details. However, there are some places where the description could be further amend it.
For this please check the detailed comments.
Page 2, line 71‐78: the reviewers give almost always only remarks to get the authors improve the manuscript. However, this paragraph reads fluently and explains with an ease why selenium rich apples should be part of scientific interest. It deserves to be said
,Page 2, line 84: EFSA assesses the submitted health claims; gives scientific opinions on the reasoning, physiological and toxicological aspects and finalize it in conclusions and recommendations. Health claims are being evaluated by EFSA and its scientific panels. However, the approval is done by EU in the scope of the legislation process .DCE, is used through the manuscript either as DCE or the full name. Identify the abbreviation at the first location and used it consequently
Lines 104‐110: finally after reading conclusions it got clear what exactly the objective of the study was. It is necessary to rewrite and clearly states that the objectives of the studies are to evaluate the willingness of the consumer to buy biofortified apples by using the online survey. Furthermore this study also evaluate s the use of health claims and packaging and some other factors. Therefore rewrite this part to undoubtedly states the goal.
Line 127: Is Respondi AG a company that performed survey. Or only recruiting the respondents
Line 138: Add the abbreviation here since DCE is used throughout the manuscript, but this is the place where the methodology is explained. And check above
Line 163: it is not clear from this the illustrative choice cards are made. This gets assumed in the text on the following pages. The design include the set of assumptions and the process of establishing, attributes definitions but also the technique used to check them, in this case choice cards. Therefore needs to be clearly said.
Page 4, line 191: in the fourth hypothesis the authors addressed that plastic free alternative is “the most preferred”. How did author made a distinction between “the most preferable” and “more preferable “to describe the choice of the packaging material?
Line 207: are those part of the choice set
Line 220:Fixed order may imply the easier choice on the card , in particular when there are 20 . the respondents may tend to overlook changes made on the packaging of a less preferable location on the choice card. Unless this 20 choice card is the part of the study selection process. It confuses the reader on about the study design. Can you comment on this.
Line 224’ : something is missing ; the text shown to the respondents
Line 226‐227:This sentence does not belong in the manuscript. It sound commercial and is odd to read it in the report of the scientific results
Line 229: the companies are launching Selstar. It is confusing. Is this study meant as a pre-launch exploration or as an evaluation of the consumer willingness. In case of first; it only matters as an observation and it would be nice to know what is the percentage of bio fortified apples currently on the market. In case of second; it would be needed to amend the sentence
Line 229‐231: although the question gives an idea of your work and buildup of the survey; unless it is not part of the methodology it should be rewritten. It is no that the reader has to imagine and answer the questions. Report the methodology and the results.
Line 232: is the choice shown on figure 1 one optimal choice set or is this the final choice set that was shown to the respondents The choice set is ambiguous: there are three different apple variety; the claims are different ; the prices in unfavorable/favorable …But it looks realistic and market representative.
Line 247: it is a rather peculiar decision to put a new beneficial apple variety into a plastic packaging with an assumption that an contentious consumer would try to avoid plastic packaging. Can you comment on this?
Lines 253: measured instead of measures
Line 253: at this point it is confusing how the respondents were tested for their willingness to buy and being able to evaluate the attributes mentioned in the equation hereafter. This should be described in the section 2.1 survey design Line 284: how does this population distribution aligns with the German population distribution; is it that only one fifth of German population has children; and is it that almost 40% of German population is a family of two. It is notable to comment since the authors claim in the abstract that a German consumer has affinity for iodine supplemented apples. Can you comment on this.
Table 5: the factor loadings in the table is shifted
Table 5: it appears from the study that plastic free packaging is one of the important factors to choose for buying. What was the impact on the Selstar packaging?
Table 5; what is the reason to include “popular aspects variety ”Elstar” into practicability factor? Is this practicability factor, part of the study or the choice? Could it be that the wording should be changed. Give an explanation and amend where necessary.
Line 3521/ regarding my previous comment given for table 5: it is now clear what this factor stands for. However it may be further renamed: since Braeburn was also shown as an popular variety but it is not include in this factor.
Line 410: this assumption is an important artefact of the study: Willingness of consumer to buy Se-biofrotified apples will be mostly influenced by their awareness, the rest will be the choice based on other attributes. If the purpose is to make more aware consumer; the statement that awareness on selenium should be further evaluated. This in turn also impact the choice to produce Se‐biofortified apples and claim it if the consumer does not care about it. The authors only comment briefly on this in line 415.
Line 432: how does this correspond to the Art.13(1) of the Commission Regulation (EU) 432/2012 of 16/05/2012? It is said that the claim may be used only for food which is at least a source of selenium as referred to in the claim SOURCE OF [NAME OF VITAMIN/S] AND/OR [NAME OF MINERAL/S] as listed in the Annex to Regulation (EC) No 1924/2006. Does this kind of apple comply with the requirements o the regulation to have the claim?
Line 457‐458: although it sounds obvious that the most popular variety would be the most preferred and with the most influences, what is the objective of this hypothesis. Is it to test whether Selstar name would be appealing, or that Pink Lady Seplus or Pink LadyS (for example) would be better?
Lines 495: it is apparent that the consumer did not take as an important factor the biofortification. Can you comment on the willingness of no‐ food supplements consumers?
Line 517: the objective of the study appears to be the evaluation of the current willingness to buy unknown product which is enriched by biofortification. This should be correctly and un ambiguously described in the introduction
Reviewer 3 Report
Selenium as an element has been described very poorly.
The paper is dense, sometimes unconnected, and rather repetitive in places (e.g. there are multiple sections that deal with Se) and the manuscript could easily be given greater clarity and focus by revising the data presented.
Nothing is known about an individual's selenium needs.
The aims of the study should be reformulated according to the research objectives and target results. The conclusions must reflect the innovation of this study and the perspectives.
The text may be handled by a native English speaker.
There are no factors influencing the absorption of selenium from the diet (apples), the authors do not write anything about it.
Why the authors did not describe anything about the influence of this element on the functioning of the thyroid gland and other human organs?
What are the negative conditions for the consumption of such apples, if it is not known whether they contain, for example, elemental selenium?
Why has the authors failed to describe the particular chemical forms of selenium that may be found in apples?
What is the point of enriching apples with selenium if the content of this element is very low in these products of plant origin?
The authors used very old literature. You should change them by giving your latest references.
What is the scientific aspect of the presented study? What is the outlook for the future?
Round 2
Reviewer 1 Report
Please, find comments on file attached

Reviewer 3 Report
The article presents nothing new all the time. The authors described the results in a very rudimentary way.
